# Pleiotropic Effects of Vitamin D in Patients with Inflammatory Bowel Diseases

**DOI:** 10.3390/jcm11195715

**Published:** 2022-09-27

**Authors:** Aleksandra Szymczak-Tomczak, Alicja Ewa Ratajczak, Marta Kaczmarek-Ryś, Szymon Hryhorowicz, Anna Maria Rychter, Agnieszka Zawada, Ryszard Słomski, Agnieszka Dobrowolska, Iwona Krela-Kaźmierczak

**Affiliations:** 1Department of Gastroenterology, Dietetics and Internal Diseases, Poznan University of Medical Sciences, 61-701 Poznan, Poland; 2Institute of Human Genetics, Polish Academy of Sciences, 60-479 Poznan, Poland

**Keywords:** inflammatory bowel disease, vitamin D, vitamin D receptor (VDR)

## Abstract

The multifaceted activity of vitamin D in patients with inflammatory bowel disease (IBD) presents a challenge for further research in this area. Vitamin D is involved in the regulation of bone mineral metabolism, it participates in the regulation of the immune system, and it is an underlying factor in the pathogenesis of IBD. Additionally, vitamin D affects Th1 and Th2 lymphocytes, influencing the release of cytokines and inhibiting tumor necrosis factor (TNF) expression and the wnt/β-catenin pathway. As far as IBDs are concerned, they are associated with microbiota dysbiosis, abnormal inflammatory response, and micronutrient deficiency, including vitamin D hypovitaminosis. In turn, the biological activity of active vitamin D is regulated by the vitamin D receptor (VDR) which is associated with several processes related to IBD. Therefore, in terms of research on vitamin D supplementation in IBD patients, it is essential to understand the metabolic pathways and genetic determinants of vitamin D, as well as to identify the environmental factors they are subject to, not only in view of osteoporosis prevention and therapy, but primarily concerning modulating the course and supplementation of IBD pharmacotherapy.

## 1. Introduction

Vitamin D has been of particular interest to researchers in recent years. It has been emphasized that it presents a multidirectional effect, in terms of both bone tissue and several extraskeletal activities, such as affecting cell proliferation and differentiation, as well as regulating gene expression. All the abovementioned functions are crucial in inflammatory bowel diseases, i.e., systemic diseases with multifactorial etiology (genetic, environmental, and immunological).

The prevalence of both Crohn’s disease and ulcerative colitis in industrialized societies is increasing, particularly among young people. Moreover, vitamin D deficiency, caused by nutritional deficiencies, vitamin and microelement malabsorption, and avoidance of sun exposure, is also a common issue, especially in patients with inflammatory bowel disease (IBD). Vitamin D metabolism and intestinal microbiota are closely intertwined, which is particularly important in this group of patients [1]. Vitamin D in patients with IBD is able to modify the function of immune cells and may affect the ongoing systemic inflammatory process that, in turn, may be reflected in disease activity vie its impact on the release of cytokines [2,3]. Furthermore, vitamin D also influences bone mineral density by regulating calcium absorption in the gastrointestinal tract and by affecting the RANK/RANKL/OPG system. This narrative review presents the current data regarding the pleiotropic effects of vitamin D in patients with IBD. The authors’ aim was to focus mainly on the incidence and prevalence of vitamin D deficiency in patients with IBD, its immune system function in this particular group of patients, and the molecular mechanisms underlying vitamin D action in IBD. Additionally, the associations of vitamin D with osteoporosis in patients with Crohn’s disease (CD) and ulcerative colitis (UC) were considered. This review also indicates that a critical element is understanding the pathogenetic mechanisms of vitamin D action, which may help to develop new therapeutic opportunities for IBD patients. Additionally, the role of vitamin D in patients with IBD has important clinical implications. An important aim in the treatment of IBD patients is to achieve not only clinical remission, but also endoscopic and microscopic remission, which may be related to numerous functions of vitamin D, particularly its impact on the immune system [4]. It should be emphasized that clinicians worldwide have been focused on developing the best and most effective treatment regimens [5]. The basis for such research are experimental studies, systematic reviews, and reviews of the literature; therefore, reviews of the literature that systematize the current state of knowledge are of great significance.

## 2. Vitamin D

Vitamin D, which serves as a prohormone, is a fat-soluble vitamin in two forms: ergocalciferol (vitamin D2) and cholecalciferol (vitamin D3). The primary sources of vitamin D in the body include dermal synthesis induced by sunlight and dietary intake [6]. Initially, previtamin D is modified to 25-hydroxyvitamin D3 (25(OH)D) in the liver by the 25-hydroxylases, CYP2R1 and CYP27A1, and further activated in the kidney by the 1-α-hydroxylase CYP27B1 [2]. On the basis of 25(OH)D concentration, it is possible to diagnose severe vitamin D deficiency, as well as deficiency, suboptimal, optimal, high and toxic concentrations of vitamin D may be diagnosed (Table 1) [7].

### 2.1. Vitamin D Metabolism

Vitamin D is transported to the liver in order to be metabolized into 25(OH)D by 25-hydroxylase (CYP2R1); 25(OH)D is the inactive form of vitamin D that is subsequently converted in the kidneys by the 1-α-hydroxylase enzyme (CYP27B1) into its active form 1,25(OH)2D. This process, in turn, is closely regulated by the parathyroid hormone (PTH), calcium and phosphate levels, and the fibroblast growth factor.

Then, 25(OH)D and 1,25(OH)2D are both degraded by 24-hydroxylases (CYP24A1) as a feedback loop negatively induced by 1,25(OH)2D. This results in calcitriol and calcitroic acid production, which are excreted in the urine or in the bile through the intestine. In vitro, calcitroic acid in higher concentrations has also been reported to bind the vitamin D receptor (VDR) and affect gene transcription. Although these studies were not conducted on the basis of intestinal tissue [8], the presence of CYP27B1 and CYP24A1 hydroxylases in the intestinal epithelial cells, immune cells, and other tissues has led to the hypothesis that local autocrine and paracrine functions may affect intestinal vitamin D signaling [8,9]. The scheme of vitamin D synthesis is presented in Figure 1.

### 2.2. Natural Sources of Vitamin D

#### 2.2.1. Food Sources of Vitamin D

The primary food sources of vitamin D include fish oil (e.g., sardines and salmon), cod liver oil, liver and organs, and egg yolk, although vitamin D has also been found in certain mushrooms (e.g., shiitake). Generally, animal products contain cholecalciferol [10], whereas mushrooms and yeast are rich in ergocalciferol. Nevertheless, non-animal vitamin D sources may be essential for individuals following a vegetarian diet [11]. Therefore, it is worth bearing in mind that some algae, such as chlorella, are also sources of ergocalciferol [12]. Moreover, Kuhn et al., presented interesting results of their study, as they reported that cocoa and cocoa-based food (e.g., chocolate) were also sources of vitamin D [13]. However, the bioavailability of ergocalciferol was found to be lower as compared with cholecalciferol [14].

Nevertheless, vitamin D content in various products is low. For instance, in the UK-based McCance and Widdowson’s Food Composition database, the percentage of products containing vitamin D above 10 µg/100 g is only 1%. Thus, in order to prevent vitamin D deficiency, in some countries traditional foods (e.g., milk and margarine) are fortified with vitamin D [15]. However, vitamin D content in animal products may be further increased by adding vitamin D to animal feed [16].

Nearly 80% of dietary vitamin D is absorbed in the jejunum and the terminal ileum. It is subsequently released from the enterocytes through the formation of chylomicrons and transported through the lymphatic system to the muscles and adipose tissue. Vitamin D as a prohormone undergoes activation [17].

#### 2.2.2. Environmental Sources of Vitamin D

Ultraviolet-B (UVB) radiation (with energy of 290–315 nm) is absorbed in the epidermis by 7-dehyrdrocholesterol and subsequently converted to previtamin D3, isomerizing into vitamin D3 [18]. Afterwards, vitamin D is metabolized in the liver and kidneys into the circulating form, 25-hydroxyvitamin D, and into the biologically active form, 1,25-dihydroxyvitamin D [19]. Although vitamin D can be provided orally through diet or supplementation, sun exposure is usually considered to be its major source. However, several factors, such as zenith angle, age, skin pigmentation, season, time of the day, clothing, air pollution, sunscreen use, and the sun exposure behaviors, are essential in the proper synthesis of vitamin D. In fact, they may significantly affect both the status of vitamin D, as well as the need for its supplementation [20]. Firstly, due to the absorption in the ozone layer, only around one percent of UVB radiation reaches the surface of the Earth. Moreover, during autumn and winter, in regions above or below 33°, solar UVB radiation is insufficient for vitamin D synthesis, and supplementation is generally required. Secondly, air pollution can also impair the absorption of solar UVB radiation [21]. Generally, higher 25(OH)D concentrations are observed during the summer season, although there are several aspects that need to be addressed in order to provide effective vitamin D synthesis [22]. It is generally accepted that higher concentrations of 25(OH)D are synthesized when a larger body surface is exposed to UVB radiation (for a minimum of 15–30 min per day from 10 a.m. to 3 a.m.), for example, about 18% of the body surface (uncovered forearms and partially legs) should be exposed for proper vitamin D synthesis [7]. Although a slight increase in the serum concentrations of 25(OH)D has been observed when only the face and hands (5% of the body surface) were exposed, it is probably insufficient in preventing vitamin D deficiencies [23,24]. Skin type is also vital; according to the Fitzpatrick skin phototype, individuals with darker skin are at an increased risk of vitamin D deficiency. Furthermore, although sunscreens are essential in preventing skin cancers, they can also reduce skin synthesis by around 90%. However, in real-life observational studies, the association between sunscreen use and lower vitamin D concentrations has not always been confirmed [25]. It is important to remember that the use of sunscreen should not be discouraged in order to provide a proper amount of vitamin D, particularly among the high-risk group of skin cancers. Moreover, an increased ultraviolet A protection factor (UVA-PF) in SPF improves the production of vitamin D. Finally, lower concentrations of vitamin D have also been observed among older individuals, which have mostly been associated with skin aging and less outdoor activity [26]. Additionally, it is vital to bear in mind that outdoor activity is also essential for proper skin synthesis of vitamin D, as higher physical activity has also been associated with higher vitamin D concentrations in the general population [27]. A summary of the potential determinants of vitamin D status is presented in Figure 2.

## 3. Inflammatory Bowel Diseases

Inflammatory bowel diseases comprise a group of chronic inflammatory conditions of the gastrointestinal tract, which include Crohn’s disease and ulcerative colitis [28]. Inflammatory lesions in the course of Crohn’s disease may cover the entire gastrointestinal tract, from the oral cavity to the anus, involving all layers of its wall. In contrast, inflammatory lesions in patients with UC involve the mucosa, where a characteristic feature is the continuity of the lesions, which start in the rectum [29]. General symptoms of inflammatory bowel disease include malaise, weight loss, subfebrile body temperature, or fever [30]. Moreover, alternating phases of exacerbation and remission are characteristic for the course of both Crohn’s disease and ulcerative colitis. It is vital to bear in mind that inflammatory bowel diseases do not affect only one organ, and thus, they should be treated as systemic diseases with several extraintestinal manifestations.

Inflammatory bowel diseases are found globally. As pointed out in a systematic review by Molodecky et al., the highest incidence and prevalence has been reported in Western countries, particularly, in northern Europe and Canada [31]. It is estimated that the disease may affect up to about 3 million people in Europe, where the annual incidences of IBD are 24.3 cases/100,000 for ulcerative colitis and 12.7 cases/100,000 for Crohn’s disease [32,33]. Moreover, according to Burisch and Munkholm, the prevalences of ulcerative colitis and Crohn’s disease have increased in Europe from 6.0/100,000 individuals per year for UC, and 1.0/100,000 persons per year for CD in 1962 to 9.8/100,000 persons per year and 6.3/100,000 persons per year in 2010, respectively [34]. Moreover, according to the available data, in 2017, 6.8 million individuals suffered from IBD worldwide. In fact, the IBD prevalence rate (number of cases per 100,000 persons) increased in the period between 1990 and 2017 (from 79.5 to 84.3). Simultaneously, the number of deaths due to IBD decreased (from 0.61 to 0.51 per 100,000 persons) [35]. It is estimated that between 2025 and 2030, 1% of the UK population will suffer from IBDs [36]. Furthermore, in Europe, an East-West gradient in the incidence of IBDs has been observed [37]. However, recently an increase in new diagnosed cases have also been observed in the East European countries [38].

The main factors contributing to an increase in the incidence of IBDs include socioeconomic trends and the resulting lifestyle changes, dietary habits, hygiene standards, and other factors comprising the so-called “Western lifestyle” [39]. Crohn’s disease occurs most frequently in young people between 16 and 25 years of age. The disease onset is usually in early adulthood (between 25 and 35 years of age), although even 20–25% of cases are observed in childhood [40].

The etiology of IBDs has not been entirely determined, although the researchers claim that it involves such factors as inadequate regulation of the immune system (abnormalities in the regulation of T-helper lymphocytes—Th), genetic and environmental factors (bacteria comprising the intestinal flora, diet, chemical compounds contained in food and the human environment, as well as smoking) leading to the chronic inflammatory process of the alimentary tract [33,41,42]. In terms of immunological factors, cytokines seem to be a particularly interesting element. In fact, the affected gastrointestinal mucosa show an increase in the number of monocytes and activated macrophages, which are the source of interleukins initiating and sustaining inflammation, as described in the subsequent section of the paper.

### Vitamin D Deficiency in the General Population and in IBD Patients

Vitamin D deficiency appears to be a worldwide problem. According to estimates, nearly one billion people worldwide suffer from vitamin D deficiency [43]. A meta-analysis by Hilger et al., that included 168,389 participants from 44 countries found vitamin D levels below 30 ng/mL in 88.1% of subjects, whereas 37.3% of subjects presented vitamin D levels below 20 ng/m [44]. In the United States, 25(OH)D levels below 20 ng/mL were observed in 36% of the healthy population aged 18–29 years, and this percentage increased to 41% in those aged 49–83 years [45]. Moreover, low vitamin D concentrations (<20 ng/mL) have also been reported among residents of the sunny regions of the United Arab Emirates, India, and Turkey [46]. Interestingly, vitamin D levels below 30 ng/mL were also found in 79.6% of postmenopausal women in the European population [47].

A group particularly prone to vitamin D deficiency are patients with inflammatory bowel diseases. According to the available data, 22–70% of patients with Crohn’s disease and about 45% of patients suffering from ulcerative colitis show vitamin D deficiency [48,49,50]. Low vitamin D levels have also been found in patients with newly diagnosed disease. Vitamin D levels < 10 ng/mL have been reported in up to 25% of patients with Crohn’s disease [51]. Nutritional deficiency in patients suffering from IBDs may stem from an insufficient intake, higher demand, or malabsorption [52]. Therefore, it is vital to bear in mind that vitamin D deficiency constitutes a common issue among patients suffering from IBD, particularly from CD. Additionally, a deficiency of vitamin D is also associated with disease activity [53]. Li et al., reported that the serum concentration of vitamin D was lower in CD and UC patients than in healthy subjects [54]. According to the studies, a standard supplementation dose was not sufficient for patients with IBD [55]. Moreover, sun exposure was an important factor affecting vitamin D status, as the number of patients complaining of IBD increased by 50% at the end of winter [56]. Importantly, vitamin D levels may also have been affected depending on the disease phase. During the moderate and severe phases of UC, serum vitamin D levels were lower than in the mild phase and in remission. In terms of CD, vitamin D levels were decreased in the moderate and severe phases as compared with those of patients in remission [57]. Additionally, vitamin D deficiency is linked to CD activity [58], and Scotti et al., also found an association between vitamin D status and IBD activity [59].

## 4. Genetic Determinants of Vitamin D Activity in IBDs

IBDs are associated with microbiota dysbiosis, abnormal inflammatory response, and micronutrient deficiency, including vitamin D hypovitaminosis. Thus, vitamin D deficiency may constitute a triggering factor and a consequence of IBD. Serum 1,25(OH)2D is involved in immune cell differentiation, gut microbiota modulation, gene transcription, and barrier integrity. Furthermore, it has additional vital roles in the intestines, including maintaining the integrity of mucosal tight junctions, enhancing folate absorption, and activating cytochrome P450 3A4 expression [60]. Several epidemiologic studies have reported low circulating serum levels of 25(OH)D in IBD patients and its relationship with IBD-related hospitalizations and surgeries, as well as inadequate responses to TNF-α inhibitors [2,58,61,62,63].

It has been demonstrated that VDR showed extensive interactions at the genomic level and that VDR binding depended on the cell type. However, only one of these studies was performed on a cancer cell line delivered from the colon, which might not have been the most suitable model for VDR genome occupancy in IBD [64].

Binding of 1,25(OH)2D to VDR leads to hetero-dimerization with the retinoid X receptor (RXR), as well as with other co-factors. Subsequently, the VDR complex translocates to the cell nucleus where it binds to vitamin D response elements (VDREs), mainly present in promoters of target genes, and either promotes or suppresses gene transcription [65]. In addition, VDR signaling regulates gene transcription by direct interaction with other regulatory protein/transcription factors, including nuclear factor kappa-light-chain-enhancer of the activated B cells (NF-kB) which induces the expression of the gene encoding antimicrobial peptide defensin β2 (DEFB2/HBD2) [66]. Furthermore, 1,25(OH)2D/VDR acts VDRE-independently as a regulator of Ca2+ and Cl− transporters [67].

A substantial genetic component predisposes to the development of UC and CD, and a number of single nucleotide polymorphic (SNP) variants are common to both diseases, hence, resulting in several pathways potentially similar [68]. Hormonal vitamin D, 1,25(OH)2D, substantially stimulates the expression of the NOD2 (CARD15/IBD1) gene (nucleotide-binding oligomerization domain-containing protein 2 gene) and protein in the primary human monocytic and epithelial cells [68]. However, vitamin D deficiency, as well as VDR deficiency or dysfunction, may also lead to IBD susceptibility.

In the studies concerning cell cultures obtained from biopsies of healthy colon tissue treated with 1,25(OH)2D, Alleyne et al., identified 465 upregulated and 417 downregulated genes across the entire genome [69]. Additionally, when intestinal epithelium from a normal colonic mucosa was tested, 182 loci were associated with the genes expressed following 1,25(OH)2D treatment. Interestingly, the highest association was found for CYP24A1, TRPV6 (transient receptor potential cation channel subfamily V member 6), and CD14 [70]. Moreover, a recent study by Kellermann et al., aligned a list of IBD risk loci (over 350 possible IBD risk genes) [71] with a list of 1,25(OH)2D-dependent genes expressed in colonic tissues [69,70]. They also identified 49 IBD risk genes regulated by vitamin D signaling, 24 of which were reported to be upregulated, and 25 downregulated.

### Vitamin D Receptor and Vitamin D Binding Protein

The VDR regulates the biological action of active vitamin D. The interaction between vitamin D and VDR is involved in the genetic, environmental, immune, and microbial aspects of IBD [72]. VDR is present in several organs and cell types, including the immune system and intestines, where it functions as a transcription factor in the cell nucleus, which controls gene expression. Moreover, intestinal VDR expression is inversely correlated with the severity of inflammation in patients suffering from IBD [73].

The human VDR is a known IBD risk gene encoding six domains of the VDR protein. Data demonstrate that VDR plays a vital role in the epithelium, which in turn suggests that it either protects against the induction of colitis, or enables effective damage repair [74]. In contrast, a mouse study indicated no effects of conditional deletion of VDR in the epithelium on the severity of chemically induced colitis, although a loss of VDR expression in macrophages and granulocytes increased mucosal proinflammatory cytokine expression [75]. Gallone et al., reported that certain genetic variations seemed to alter the binding affinity of VDR for 1,25(OH)2D by altering DNA-binding affinity to other subunits of larger bio-complexes, or factors inhibiting RXR-VDR heterodimerization. These results, in turn, confirm that the altered VDR binding, and consequently the altered vitamin D levels, partly account for an increased risk of autoimmune and other complex diseases [76].

Moreover, single-nucleotide polymorphisms in the human VDR gene were reported to be associated with an elevated susceptibility to IBD [77,78]. The four most frequently studied are common VDR polymorphisms recognized by restriction enzymes: rs7975232 (c.1025-49 G > T, ApaI) and rs1544410 (c.1024 + 283 G > A, BsmI) at the 3′ flanking end of intron 8, rs731236 (c.1056 T > C, p.Ile352=, TaqI) at the 3′ flanking end of exon 9, and rs2228570 (c.2T > C, p.Met1Thr, FokI) at the 5′ end of exon 2 [79,80]. Nevertheless, studies investigating these polymorphisms have reported different variants to be associated with an increased risk of IBD [80,81,82]. Furthermore, correlations between the rs2228570 (FokI) polymorphism and low levels of serum 25(OH)D have been observed in UC patients [83]. Interestingly, some genetic variations located preferentially within the enhancer regions have also been indicated to alter the binding affinity of VDR, and as being significantly associated with immune and inflammatory diseases [76].

Free 25OHD, mainly linked to the GC vitamin D-binding protein (DBP), is transported in the blood. DBP encoding gene, i.e., GC gene (DBP, VDBP, HGNC: 4187) polymorphism leads to differences in affinity for both 25(OH)D and 1,25(OH)2D. The GC gene sequence variants were correlated with DBP protein and circulating 25(OH)D concentrations. Moreover, higher DBP concentrations were found to be positively correlated with the exacerbation of IBD, and inversely correlated with inflammation in pediatric IBD patients [67,68].

Comparing the frequency of two SNPs in the GC gene between IBD patients and controls, Eloranta et al., found that the haplotype consisting of wild type alleles, 416 Asp and 420 Lys, was less frequent in the IBD cases [60].

## 5. Vitamin D and the Immune System in IBD Patients

Vitamin D is an essential regulator of calcium-phosphate metabolism, although it also performs immunomodulatory functions [84]. It has been shown that the vitamin D receptor and vitamin D-metabolizing hydroxylases are expressed in various immune cells, hence, vitamin D affects both innate and acquired immunity [85].

As already mentioned, a Th1 response is predominant in Crohn’s disease, whereas a Th2 response is dominant in patients with ulcerative colitis [86]. Additionally, an essential role in the pathogenesis of the inflammatory response in IBD has been attributed to Th17 cells, which produce IL-17A (interleukin 17A) and proinflammatory cytokines [87]. These cells, in turn, can additionally transform into Th1 or T-reg cells. In fact, vitamin D has an inhibitory effect on T-lymphocyte proliferation, which also alters the cytokine profile, and a preferential shift from Th1 and Th17 to Th2 profiles is observed, which results in blocking the production of IL-2, IFN gamma and TNF-α, IL-17 and IL-21 and increasing the expression of IL-4, IL-5, IL-9, and IL-13 [87,88,89,90,91,92,93]. Moreover, vitamin D also affects the inhibition of the inflammatory process by stimulating the differentiation of Treg lymphocytes, both in a direct mechanism and through cooperation with antigen-presenting cells [94].

It is also worth noting that vitamin D affects the regulation of B lymphocyte activity. The inhibition of their activity and maturation towards plasma cells contributes to a decrease in autoantibody production, which, in turn, may lead to a decreased risk of IBD. Additionally, 1,25(OH)2D stimulates B lymphocytes to produce anti-inflammatory cytokines, mainly IL-10 [95]. A decrease in B lymphocyte proliferation induced by calcitriol is also associated with the induction of apoptosis [96]. Furthermore, vitamin D also impacts dendritic cell maturation, inhibits the expression of MHC II major tissue compatibility system antigens, and CD40, CD80 and CD86, leading to improved immune tolerance [97]. Vitamin D also indirectly affects the shift of the immune response towards Th2 by stimulating dendritic cells to secrete IL-10 and by inhibiting IL-12 secretion. Additionally, vitamin D stimulates the innate response by enhancing macrophage function, including chemotactic and phagocytotic properties, and by producing antimicrobial peptides, i.e., cathelicidins. A critical pathway regulating tissue damage and repair in the inflammatory processes occurring in the intestines of IBD patients is the Wnt/β-catenin pathway [98]. Clarifying the reciprocal action of Wnt ligands and cytokines could potentially help identify new treatment options for chronic colitis and other inflammatory disorders. Vitamin D results in the inhibition of Wnt pathway signaling [99], which occurs through various mechanisms in a VDR-dependent manner. There is a VDR–β-catenin interaction and stimulation of nuclear β-catenin export. In addition, 1,25(OH)2D also leads to an increase in Dickkopf-related protein 1 (DKK1), which also acts as an inhibitor of Wnt signaling [100]. In addition, the active form of vitamin D increases cystatin D levels, which, in turn, also inhibits the Wnt pathway.

Due to the significant effects of vitamin D on the immune system in patients with IBD, it has been suggested that 25(OH)D levels are associated with disease activity in this patient group. According to a meta-analysis by Gubatan et al., who comprised 27 studies including 8316 patients with IBD, vitamin D levels were associated with an increase in disease activity, relapses, and reduced quality of life in these patients [101]. Moreover, Utilsky et al., found that, in CD patients, vitamin D deficiency was associated with a decreased health-related quality of life (HRQOL), as well as with increased disease activity assessed by using the Harvey Bradshaw scale [58]. Similarly, Harries et al., also indicated an association between low vitamin D concentrations and greater disease activity in patients with CD [102]. A study by Ananthakrishnan et al., which involved 3217 patients with IBD, revealed that, in patients with CD, 25(OH)D levels < 20 ng/mL were associated with an increased risk of surgery and hospitalizations related to the underlying disease as compared with patients with 25(OH)D levels ≥ 30 ng/mL. Similar observations were found in patients with UC [103]. A study by Cantorn et al., conducted in IL-10-deficient mice (IL knockout) appeared to provide substantial evidence for the association between the inflammatory bowel disease activity and vitamin D. Additionally, the studies demonstrated that the administration of 1,25(OH)2D or cholecalciferol was correlated with an improvement in intestinal inflammation, with a dose-response effect [104]. The observations from a study by Zhu et al., conducted in an animal model were particularly relevant, as they found that an intervention involving vitamin D administration was associated with a reduction in TNF-related gene expression in the intestines [105].

In a study conducted by Ananthakrishnan et al., on 2809 patients with IBD, reduced plasma 25(OH)D levels were associated with an increased risk of cancer, particularly colorectal cancer [106]. Vitamin D may also show preventive properties against colorectal cancer associated with chronic inflammation in patients with IBD. However, the mechanisms of antineoplastic effects of vitamin D and its active form have not been fully understood, although it has been hypothesized that it affects multiple signaling pathways and regulates the expression of multiple genes. In addition, 1,25(OH)2D can inhibit tumor cell proliferation by stimulating the expression of inhibitors of cyclin-dependent kinases, such as p21, p27, and cystatin D, and by inhibiting the expression of pro-proliferative genes, including c-Myc and cyclin D1 [107]. Furthermore, 1,25(OH)2D also affects colorectal cancer cells both in vitro and in vivo through epigenetic regulation by affecting the regulation of microRNA-627, which, in turn, interacts with histone demethylase jumonji domain-containing protein 1A [108]. Additionally, vitamin D also modulates cell differentiation by affecting the expression of alkaline phosphatase, maltase, E-cadherin, and cell adhesion proteins and may enhance apoptosis [107]. One of the mechanism through which vitamin D exerts its anticancer activity is by means of the regulation of the Wnt/β-catenin pathway. Inflammation present in the intestines of IBD patients may stimulate Wnt signaling, while vitamin D may contribute to its inhibition. Therefore, it has been suggested that the underlying mechanism comprises vitamin D promotion of VDR/β-catenin binding in cancer cells, which contributes to the inhibition of the nuclear translocation of β-catenin [109].

Currently, there is an on-going debate as to whether vitamin D deficiency is a cause, or a consequence of IBD. Some researchers have referred to a link between the highest incidence of IBD and the latitude of areas with the least amount of sunlight [110]. As outlined in the paper, some studies have suggested an association between low vitamin D levels and increased disease activity. Moreover, it is worth noting that genetic factors may also have influenced the serum vitamin D levels in the patients studied [111]. It has been postulated that gene polymorphisms of vitamin D metabolic pathways may influence the body’s vitamin D supply status [112]. A prospective cohort study by Ananthakrishnan et al., which involved 72,719 women enrolled in the Nurses’ Health Study, found that higher predicted plasma 25(OH)D concentrations significantly reduced the risk of CD and non-significantly reduced the risk of UC in women [113]. Therefore, as it has already been emphasized in the presented paper, it seems essential to identify and understand the metabolic pathways and molecular basis of vitamin D action in IBD patients, which requires further research.

### Proinflammatory Cytokines

A key pathophysiological element of inflammatory bowel disease is the cytokine response, which is responsible for inflammation regulation, evolution, and occurrence. It consists of two different levels of cellular response. The first comprises the process of cytokine differentiation from Th1 and Th17 cells, where the central cytokines are IFN-γ and IL-2 and IL-17. These cytokines, in turn, support the cellular response and are particularly involved in Crohn’s disease, in which IFN-γ plays a significant role [114]

Conversely, in ulcerative colitis, a Th2-like differentiation process is most relevant, in which NKT cells that produce IL-13 and IL-5 are crucial.

These aforementioned IBD-specific cytokine patterns contribute to the second level of cytokines response, involving Th1, Th17, and Th2 differentiation processes, conditioning the humoral response and mediating the inflammatory process. These cytokines are TNF-α, IL-1β, IL-6, IL-4, IL-5, and IL-10 [29]. In fact, studies by Hummel et al. (2014) have shown that IL-6 and TNF-α play a critical role in inflammatory bowel disease. They demonstrated that treatment with TNF-α and IL-6 led to a decreased expression of the vitamin D activating enzyme CYP27B1. Based on this observation, they concluded that the presence of proinflammatory cytokines might impair the activation of 1,25(OH)2D, limiting its anti-inflammatory action [115]. Recently, several studies have investigated the relationship between IBD risk genes and vitamin D signaling in both mice and in vitro cell models. The most studied genes were VDR, CLDN2, ATG16L (autophagy-related 16-like 1), PTPN2, and NOD2. VDR-signaling through its effects on ATG16L1 and NOD2 proteins is implicated in autophagy and protects against the apoptosis of intestinal epithelial cells (IECs) disabling the NF-κB activation.

Studies regarding the intestinal epithelial cell line Caco-2 have shown that 1,25(OH)2D regulates the expression of zonula occludens 1 (ZO-1) and occludin, which are vital compounds of the intestinal barrier [116]. These proteins are, in fact, structural components of the apical junctional complex (AJC), controlling the paracellular transport of micro- and macromolecules. An imbalance in this intercellular binding, facilitated by a dysbiotic intestinal community, results in increased barrier permeability, allowing bacterial metabolites to reach the basal intestinal layer and stimulating local inflammatory processes [117]. One of the critical transmembrane proteins in the AJC is E-cadherin, connecting this complex to the actin cytoskeleton. Interestingly, Meckel et al., reported that E-cadherin was downregulated in UC patients with decreased 25(OH)D serum levels as compared with patients with higher 25(OH)D levels [90].

## 6. Vitamin D, Microbiome, and IBD

The intestinal microbiota is a group of microorganisms (microbiome), mainly bacteria, that form a complex ecosystem in the gastrointestinal tract. The human gastrointestinal microbiota is composed mainly of bacteria, although it also contains Archaea, microeukaryotes, and various viruses, mainly bacteriophages. Nearly 2000 metagenomic species of bacteria have been described, which are distributed throughout the gastrointestinal tract. Interestingly, the density of the populations is not constant, and the highest density has been observed in the large intestine [118]. The intestinal microbiota perform a number of functions, including metabolism of nutrients, production of short-chain fatty acids and vitamins, as well as participation in the fermentation processes of dietary fiber and affecting the proper functioning of the intestinal barrier [72].

*Firmicutes*, *Bacteroidetes*, and *Proteobacteria* are the most abundant bacteria in the human intestinal microbiome composition [119]. However, the composition of the microbiome in patients with IBD differs significantly as compared with healthy individuals. In fact, a significant increase in *Proteobacteria* (*Enterobacteriaceae*, *Escherichia coli*) has been described in IBD patients. Moreover, the composition of the microbiome may also differ between CD and UC patients. It has been shown that, in patients with CD, the number of bacteria of the genus Firmicutes, in particular *Faecalibacterium prausnitzii*, was significantly reduced, whereas the numbers of bacteria of the genus *Proteobacteria* and *Bacteroidetes* was increased. In contrast, in patients with ulcerative colitis, the number of butyrate-producing bacteria in the intestinal microbiota was significantly reduced [72].

The multipotential activity of vitamin D has been known for several years, however, its effect on autoimmune diseases has not been fully understood. There are reports according to which vitamin D may regulate gastrointestinal inflammation by binding to the intracellular receptor VDR, affecting the transcription of relevant genes. Nevertheless, the main effect of vitamin D on the intestinal microbiome is altering the transcription of the immune proteins as cathelicidin and defensin and enhancing intestinal barrier function [85]. In turn, cathelicidin and defensin, also possess VDR binding sites in their structure. Stimulation of the synthesis of these proteins occurs through toll-like receptors (TLRs) located on the surface of macrophages and monocytes, what is a signal to increase CYP27B1 expression, catalyzing the conversion of calcidiol to calcitriol which is the active form of vitamin D. The resulting complex of calcitriol and the VDR receptor is a transcription factor that affects genes encoding cathelicidin and defensin [120]. In addition, TLR4 signaling presumably influences intestinal flora by altering gastrointestinal motility, resulting in the removal of pathogens and increased intestinal commensal flora [121].

Vitamin D increases transepithelial resistance and decreases intestinal mucosal permeability, thus, preventing endotoxins and LPS from entering the bloodstream. Moreover, vitamin D deficiency leads to NF-kB activation and increases the transcription of cathelicidin and defensin [68]. The loss of microbiome control by the immune system leads to diffusion of inflammation within the intestinal wall. It has been observed that the severity of colitis was associated with increased levels of the normal gastrointestinal commensal bacteria in mice receiving a vitamin D deficient diet. This also emphasizes the significance of vitamin D in maintaining the integrity of the intestinal barrier [122]. Therefore, the relationships among the intestinal microbiome, vitamin D, and the inflammatory process in the intestine requires further analysis. This appears particularly relevant in view of the fact that vitamin D deficiency results in dysbiosis of the gut microbiome and is associated with the development and severity of IBD [113], as well as with an increased risk of malignant transformation of inflammatory bowel disease [106]. It is worth noting that the decreased amount of vitamin D that is commonly found in patients with IBD stems from reduced oral absorption of vitamin D [123]. This phenomenon is associated with impaired bile acid circulation and malabsorption of fat-soluble vitamins. In addition, resection of the terminal ileum may also affect vitamin D absorption [124]. Therefore, vitamin D supplements have been suggested as the adjunctive treatment in inflammatory bowel disease [125,126].

Vitamin D is also involved in detoxifying lithocholic acid (LCA) produced by the intestinal microbiome, which may adversely affect intestinal and liver cells [127]. Kong et al., demonstrated that VDR−/− mice were more sensitive to intestinal mucosal damage and to the consequences of this damage induced by dextran sulfate sodium (DSS) as compared with VDR+/+ mice [128]. Decreased 1,25 (OH)2D production or VDR expression may lead to intestinal inflammation and increased colonization by *Proteobacteria*. *Chlamydia trachomatis* infection affects the gut microbiota and causes a decrease in VDR activity [129]. In contrast, as demonstrated in the studies, supplementation with the probiotic *Lactobacillus reuteri* increased serum 25(OH)D levels [130]. Moreover, VDR deletion in intestinal epithelial cells leads to defective autophagy in colitis, observed in inflammatory bowel disease. Interestingly, there is a crucial correlation among autophagy, VDR, and the gut microbiome, which affects intestinal homeostasis and plays a significant role in the pathophysiology of inflammatory bowel disease [131]. All the abovementioned data indicate that vitamin D is an essential factor affecting the composition of the intestinal microbiome and the inflammatory process that occurs in inflammatory bowel disease.

## 7. Vitamin D and Osteoporosis in Patients with IBD

Osteoporosis is defined as a systemic skeletal disease with low bone mineral density and a disruption of bone tissue microarchitecture, leading to an increased risk of fracture [132]. The World Health Organization (WHO) and the National Osteoporosis Foundation consider the Dual-Energy X-ray Absorptiometry (DXA) test as the gold standard for diagnosing osteoporosis. The test determines bone mineral density in the lumbar spine and in the femoral neck, and the diagnosis is established on the basis of the T-score (standard deviation number, where peak bone mass is the reference point). According to these guidelines, osteoporosis has been defined as a T-score ≤−2.5 SD, whereas osteopenia has been diagnosed with a T-score between −1 and −2.5 SD (standard deviation), and a T-score >−1 SD is considered normal [133,134]. Osteopenia and osteoporosis both represent clinically relevant issues in patients with IBD. According to the available data, their prevalence is 22–77% for osteopenia, and 17–41% for osteoporosis, depending on the studied population [135]. Adriani et al., in their research, observed osteopenia in 46% of the IBD patients studied, while osteoporosis was found in 11% of the patients; additionally, most men aged >30 years (63%) and young women (62%) presented bone mineral density abnormalities [136]. Moreover, in a systematic review by Kärnsund et al., comprising 12 studies, the prevalence of osteoporosis in the studies involving patients with CD and UC ranged between 4 and 9%. Conversely, in the studies involving patients with UC, it ranged between 2% and 9%, and in the studies consisting of patients with CD it ranged between 7% and 15% [137]. In our study, performed on patients from the Wielkopolska region (Poland), osteoporosis in the lumbar spine was found in 11.7% of CD patients, and 3.8% of UC patients. Additionally, osteoporosis was observed in the femoral neck in 5.8% and 2.9% of CD and UC patients, respectively. In turn, osteopenia was diagnosed in 35.9% (femoral neck) and 36.9% (lumbar spine) of CD patients, and in 25.7% (femoral neck) and 29.5% (lumbar spine) of UC patients [138]. It is worth noting that significant risk factors for reduced bone mineral density in patients with IBD include malnutrition, low BMI, vitamin D and calcium deficiency, hypogonadism, smoking, genetic factors, an increased systemic inflammation, and glucocorticosteroid use [139]. Recently, vitamin D has been of particular interest to researchers concerning bone metabolism in patients with IBD.

As it has already been mentioned, generalized inflammation constitutes a significant risk factor for bone mineral density loss in patients with IBD [140]. Inflammatory mediators have been demonstrated to affect osteoclasts and osteoblasts, thus, leading to impaired bone mineralization and bone mineral density loss. However, in this process, the crucial factor is the RANK/RANKL/OPG pathway [141,142], as this system accounts for the formation, activation, and survival of osteoclasts. Receptor activator of nuclear factor-B (RANK) is a protein expressed on osteoblasts and activated T-lymphocytes; it plays the role of a cytokine receptor, belonging to the TNF receptor family. The ligand for this receptor is receptor activator of NF-κB ligand (RANKL), which is expressed on the surface of T- and B-lymphocytes. The association of RANKL with RANK leads to differentiation of the precursor cells into osteoclasts, activation of osteoclasts, and regulation of their survival, which, in turn, entails bone resorption. Osteoprotegerin (OPG) is a protein produced by osteoblasts that binds to RANKL, thus, inhibiting RANKL-RANK binding, and therefore, inhibiting osteoclast activation [143]. Furthermore, the RANK/RANKL/OPG system is modulated by several factors, such as hormones, cytokines, drugs, and vitamin D, which is particularly important in the pathogenesis of osteoporosis in patients with IBD. Vitamin D stimulates RANKL expression and inhibits OPG release by preosteoblasts, hence, contributing to the stimulation of bone remodeling. In addition, 1,25(OH)2D also affects the adhesion of osteoclast precursors to stromal osteoblasts via intercellular adhesion molecule 1 (ICAM-1) [144]. The interleukins IL-1, IL-6, and TNF-α released from the inflamed intestinal mucosa stimulate the expression of RANKL, which, in turn, leads to the stimulation of osteoclastogenesis. Additionally, TNF stimulates osteoblast apoptosis simultaneously inhibiting osteoclast apoptosis. Therefore, this molecule is the key element linking the pathogenesis of IBD to abnormal bone metabolism in the course of IBD [145]. Dadaei et al., demonstrated that vitamin D supplementation in IBD patients resulted in decreased TNF-α levels, which may have indirectly contributed to the inhibition of bone tissue destruction [146].

The Wnt/β-catenin (wingless-type like signaling) pathway also plays a vital role in the regulation of bone metabolism. Wnt glycoproteins bind to the receptor complex, which includes the receptor for very-low-density lipoproteins LRP5 or LRP 6. Activation of the signal transduction pathway leads to stabilization of β-catenin, and the effect role of the Wnt/β-catenin pathway is stimulation of bone formation. A critical regulator of the Wnt pathway is sclerostin, a glycoprotein produced by mature osteocytes in response to bone-damaging stimuli [147]. Sclerostin, in turn, inhibits the Wnt pathway by binding to low-density lipoprotein co-receptors [148]. Fretz et al., showed that 1,25(OH)2D was able to increase the expression of the Wnt signaling co-regulator LRP5, which is a critical component of the Wnt signaling pathway [149]. Moreover, in stromal cells, vitamin D inhibits the expression of the important Wnt pathway antagonists, i.e., Dickkopf-related protein 1 (DKK1) and secreted frizzled-related protein 2 (SFRP2) [150]. In addition to the pathways crucial for osteoclasts, vitamin D also has a significant impact on osteoblasts. The active form of vitamin D, via mitochondrial factor inhibition, suppresses osteoblast apoptosis. In osteoblasts treated with 1,25(OH)2D, Duque et al., demonstrated a decrease in the expression of the pro-apoptotic protein Bax with a substantial increase in the expression of the anti-apoptotic protein Bcl-2 [151]. Additionally, vitamin D stimulates the production of osteocalcin, osteopontin, and bone sialoprotein by osteoblasts [152], which are associated with bone mineralization.

Another significant risk factor for osteoporosis in patients with IBD is the use of glucocorticosteroids (GCS) in the course of treatment. These drugs contribute to inhibition of the inflammatory response; nevertheless, they also negatively affect bone tissue, and their administration results in a significant decrease in bone mineral density and an increased risk of fracture, since they stimulate osteoblast apoptosis and decrease the number of osteoblast precursors. Interestingly, the Wnt/β-catenin pathway is partially involved in this process along with increased expression of sclerostin, which antagonizes Wnt signaling. The Wnt/β-catenin pathway, as a pathway which promotes osteoblastogenesis, may also be important in bone mineral density disorders in IBD patients. Ohnaka et al., indicated that Wnt3a increased T-cell factor (Tcf)/lymphatic enhancer factor (Lef)-dependent transcriptional activity in primary cultured human osteoblasts, whereas glycocorticosteroids were a significant inhibitor of this process. The researchers also showed that 1,25-dihydroxyvitamin D3 stimulated this activity. Hence, Ohaka et al., speculated that glucocorticosteroids inhibited the Wnt pathway in cultured human osteoblasts, partly by increasing the production of Dickopf-1 [153]. This provided a significant association of the Wnt/β-catenin pathway with osteoporosis, vitamin D deficiency, and glucocorticosteroid therapy in IBD patients, although they have stressed that these associations require further studies. In addition, glucocorticosteroids have been shown to inhibit osteoclast proliferation simultaneously resulting in an increased osteoclast activity due to an increase in RANKL and a decrease in osteoprotegerin levels [154]. In fact, Shymanskyi et al., showed that cholecalciferol supplementation with concomitant prednisolone treatment prevented changes in the RANKL/RANK/OPG system. As pointed out in their study, the administration of vitamin D led to a decrease in phosphoNF-κB p65 levels and inhibited the movement of NF-kB to the nucleus. This, in turn, resulted in a decrease in osteoclast precursors in bone marrow and in peripheral blood [155]. Through the stimulation of calcium absorption in the gastrointestinal tract and kidneys, vitamin D also has an indirect effect on bone. Calcium absorption occurs mainly in the duodenum and in the proximal part of the jejunum. This process involves the paracellular, or transcellular pathway, which is specifically regulated by vitamin D. Moreover, 1,25(OH)2D activates the calcium channels TRPV5 and TRPV6 located in the brush border, calbindin, and the calcium pump. The study by Huybers et al., was based on a murine model (TNF-α-overexpressing TNFDARE mice, “Crohn’s-like ileitis”), and demonstrated a significant reduction in the expression of TRPV6, calbindin, and a number of other factors involved in calcium absorption. In addition, these changes were accompanied by reduced trabecular and cortical bone thickness and volume [156]. In view of these data, vitamin D may contribute to maintaining normal serum calcium concentrations, and thus, ensuring normal bone mineralization in patients with IBD [157]. The mechanisms of vitamin D action on bone tissue in patients with IBD are presented in Figure 3.

Due to its direct and indirect involvement in a number of metabolic pathways that regulate bone tissue metabolism, vitamin D is of particular importance in patients with IBD. However, further research involving this selected group of patients is necessary, with a particular need for randomized studies on large groups of subjects. Furthermore, understanding the molecular mechanisms determining the effect of vitamin D on bone tissue in patients with CD and UC is of clinical importance, since it may improve clinical interventions.

## 8. Vitamin D and Supplementation in Patients with IBD

The importance of vitamin D in patients with IBD implies that maintaining adequate vitamin D levels in these patients constitutes an important clinical issue. However, no clear guidelines on vitamin D supplementation in patients with IBD have been developed. According to the European Crohn’s and Colitis Organisation (ECCO) guidelines concerning the management of the extraintestinal complications in IBD, a preventive vitamin D supplementation is recommended in patients receiving systemic steroid therapy, as well as in patients with osteopenia and osteoporosis. Thus, the most appropriate management seems to be evaluating vitamin D levels in patients with IBD and the supplementation/treatment of the deficiency given the guidelines for the general population [158]. Furthermore, in their cross-sectional study of 182 CD patients, Jørgensen et al., demonstrated that 25(OH)D levels were inversely correlated with the disease activity, and it was also revealed that patients who supplemented vitamin D presented with a lower Crohn’s disease activity index and C-reactive protein as compared with patients with no supplementation [159]. Additional factors which should be taken into account when considering vitamin D supplementation in IBD patients include disease activity, malabsorption, the degree of sun exposure, patient compliance, and coexistent morbidity of obesity [5]. Supplementation may provide important significant extraskeletal advantages in patients with IBD, including the modification of modifying the inflammatory response, thus, affecting the course of the disease, antitumor effects, gut microbiota modulation, and the prevention of respiratory tract infections. Vitamin D may also act as a TNF-α inhibitor, which may be important in patients undergoing biological therapies such as infliximab. In these patients, vitamin D acts synergistically with the applied treatment [160]. As pointed out by Winter et al., in their study involving 173 subjects with IBD, patients with normal vitamin D levels were more likely to achieve remission following 3 months of therapy as compared with patients with vitamin D deficiency [161]. Therefore, vitamin D supplementation may also affect the effectiveness of treatment.

Furthermore, Arihiro et al., demonstrated that the incidence of upper respiratory tract infections was significantly lower in a group of IBD patients who supplemented with vitamin D at 500 IU/day as compared with a placebo group, and this effect was most prominent in patients with low baseline 25(OH)D levels [162]. In fact, studies referring to vitamin D status and to the risk of severe acute respiratory syndrome coronavirus 2 (SARS-CoV-2) are scarce. However, IBD patients do not present a higher risk of SARS-CoV-2 infection as compared with other subjects with gastrointestinal diseases [163]. Importantly, vitamin D plays a considerable role in adaptive immunity and in regulating innate immunity, therefore, vitamin D deficiency may increase the risk of infections [164]. In fact, Scolaro et al., reported that deficiency of vitamin D was related to the laboratory and clinical activity of IBD [165], which may also lead to increased risk of infections. It is vital to note that a strong association between vitamin D deficiency and severe viral infections was found [166]. In light of the information presented above, it could be hypothesized that vitamin D may protect IBD patients from SARS-CoV-2, and that preventing vitamin D deficiency is essential for infections prevention.

Patients with inflammatory bowel disease are a group particularly susceptible to vitamin D deficiency and the associated implications. However, it is not entirely clear whether vitamin D deficiency in these patients is the cause or the result of IBD. It has been suggested that avoiding UV exposure may account for vitamin D deficiency in patients with CD and UC, which may be related to the need to stay indoors due to the disease. Another reason for avoiding sun exposure in this patient group may be excessive photosensitivity of the skin associated with the use of certain medications [1].

## 9. Summary and Conclusions

Due to its multifunctional effects, vitamin D plays a significant role in patients with IBD. Vitamin D influences the expression of known IBD risk genes and modulates immune function by influencing the activation and differentiation of immune cells. Additionally, it also affects the intestinal microbiome and maintains the integrity of the intestinal barrier, as well as contributes to normal bone mineralization. The potential therapeutic value of vitamin D in patients with IBD and, consequently, the improved clinical status of patients is also emphasized. Therefore, further research is necessary to understand, in detail, the pathways of vitamin D activity in relation to the immune system, bone tissue, and gene expression regulation. Moreover, there is also a need to establish guidelines concerning vitamin D supplementation and treatment for patients suffering from IBD. The most significant information and studies presented in the paper are showed in Table 2.

**Table 2 jcm-11-05715-t002:** The most significant information and studies presented in the paper.

Topic	The Most Significant Information	References
Vitamin D deficiency	One billion people suffer from vitamin D deficiency worldwide	[43]
	Vitamin D deficiency in IBD: 22–70% in Crohn’s disease 45% in ulcerative colitis	[48,49,50]
Vitamin D, the risk and course of inflammatory bowel disease	Higher predicted plasma 25(OH)D concentrations significantly reduced the risk of CD and non-significantly reduced the risk of UC in women	[113]
	25(OH)D levels were inversely correlated with disease activity, and patients supplementing vitamin D presented a lower Crohn’s disease activity index and C-reactive protein as compared with patients with no supplementation	[159]
	Treatment with TNF-α and IL-6 resulted in a decreased expression of the vitamin D activating enzyme CYP27B1	[115]
Genetic factors, vitamin D, and inflammatory bowel disease	49 IBD risk genes regulated by vitamin D signaling, 24 of which were reported to be upregulated, and 25 downregulated	[71]
	Single-nucleotide polymorphisms in the human VDR (vitamin D receptor) gene were reported to be associated with an elevated susceptibility to IBD	[77,78]
	Vitamin D receptor and vitamin D-metabolizing hydroxylases are expressed in various immune cells, hence, the impact of vitamin D on both innate and acquired immunity	[85]
	A decreased 1,25 (OH)2D production or VDR expression may lead to intestinal inflammation and increased colonization by *Proteobacteria*	[129]
Gut microbiota and vitamin D	Chlamydia trachomatis infection affects the gut microbiota and causes a decrease in VDR activity	[129]
	Supplementation with the probiotic *Lactobacillus reuteri* increased serum 25(OH)D levels	[130]

## Figures and Tables

**Figure 1 jcm-11-05715-f001:**
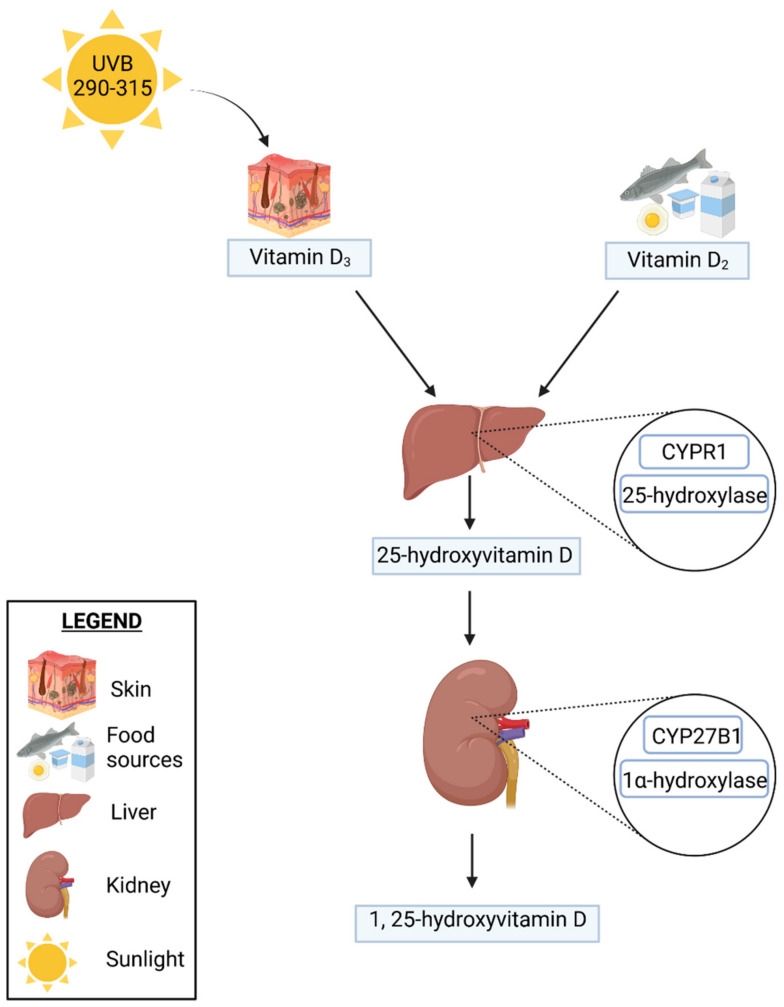
The scheme of vitamin D synthesis.

**Figure 2 jcm-11-05715-f002:**
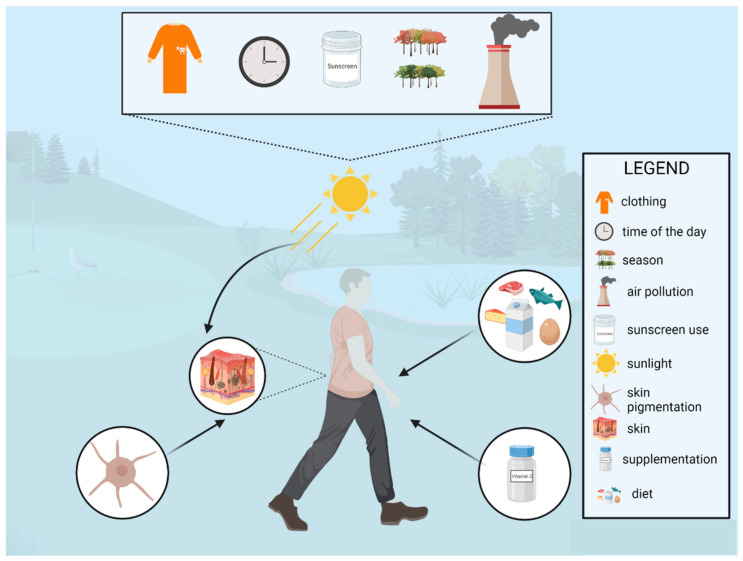
A summary of the potential determinants of vitamin D status.

**Figure 3 jcm-11-05715-f003:**
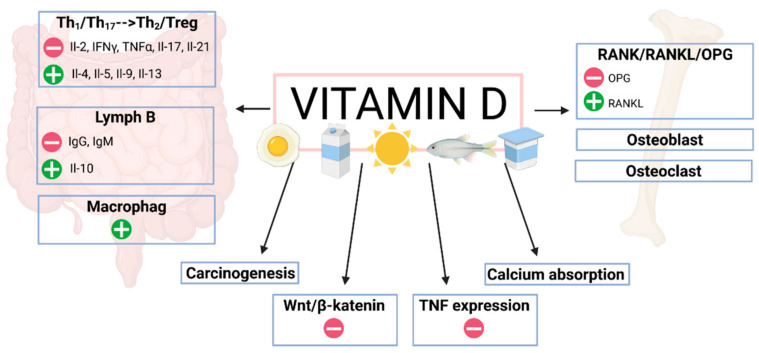
The mechanisms of vitamin D action on bone tissue in patients with IBD.

**Table 1 jcm-11-05715-t001:** Definition of vitamin D status according to the 25(OH)D concentration for Central Europe.

Interpretation	25(OH)D Concentration (ng/mL)
Severe deficiency	0–10
Deficiency	10–20
Suboptimal concentration	21–30
Optimal concentration	31–50
High concentration	50–100
Toxic concentration	>100

## Data Availability

Not applicable.

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
