# Peer review of "Pleiotropic Effects of Vitamin D in Patients with Inflammatory Bowel Diseases"

_jcm, 2022, doi:10.3390/jcm11195715_

Round 1
Reviewer 1 Report
The manuscript is well written, has scientific relevance and the figures are very well drawn.
However, I have some considerations, perhaps many of them, is a lack of attention from the authors, among them:
- Vitamin D “D” is lowercase in the lines: 80, 222, 338, 503, and 613;
- The word inflammation is duplicated on line 305;
- Bacteria species are not italicized in the lines: 493, and 495;
- And is upper case on line 662 of the conclusion;
- The abbreviation IBD is in lower case in the lines: 222, and 338;
Between lines 635 and 636, the authors say that patients with IBD who are treated with vitamin D supplementation have minor upper respiratory tract infections, according to the study of Arihiro et al. Have you found any relationship in the scientific literature regarding the coronavirus? You should broaden this discussion and if there are no works, raise hypotheses.
Reviewer 2 Report
In this review article, Szymczak-Tomczak et al summarized the current body of knowledge on the effects of vitamin D and its implications in inflammatory bowel diseases. The authors discussed the sources, metabolism, and deficiency of vitamin D, the role of vitamin D receptor in IBD, the functional role of vitamin D in the inflammatory response and regulation of cytokines in IBD, the interaction between vitamin D, microbiome and IBD, and the effect of vitamin D on osteoporosis in the setting of IBD. Overall, this review is thorough, but presents the information in a disorganized and confusing manner. In addition, this topic has been reviewed extensively in the literature. In fact, at least two review papers have been published in 2022 alone on the role of vitamin D in IBD (see https://www.ncbi.nlm.nih.gov/pmc/articles/PMC8779654 and https://www.dovepress.com/the-role-of-vitamin-d-in-immune-system-and-inflammatory-bowel-disease-peer-reviewed-fulltext-article-JIR), so this review manuscript does not seem to add much to the existing body of literature. Furthermore, spelling errors, grammatical errors, and language use should be revised extensively. My specific comments are listed below:
1. Line 141-154: the metabolism of vitamin D was discussed under subsection 3.2 environmental sources of vitamin D, which does not seem to be appropriate. I suggest that you make a new section subheading for these two paragraphs. In addition, it doesn't seem appropriate to put this content in section 3, which is titled "natural sources of vitamin D".
2. Again, section 3 is titled "natural sources of vitamin d", yet subsection 3.3 focuses on vitamin D deficiency. I suggest either changing the section 3 title to make it more encompassing or moving subsection 3.3 to another section.
3. For subsection 3.3, you discuss vitamin d deficiency in inflammatory bowel disease patients, and yet you introduce IBD in section 4. A more logical way to organize and present this content is to introduce IBD first, and then discuss vitamin D deficiency in the setting of IBD later in your review article.
4. Section 2 and 3 can be combined as these two sections are both introducing vitamin D.
5. Line 233-253: VDR and the implications of VDR in IBD are discussed here, but it would be better to move this content under subsection 5.1 which focuses on VDR.
6. Line 293-302: vitamin D binding protein is discussed under subsection 5.1 entitled “vitamin D receptor”. I suggest either changing the subsection title or making this content on vitamin D binding protein a standalone subsection.
7. I am confused as to why subsection 5.2 “proinflammatory cytokines” is listed under section 5, which is entitled “genetic determinants of vitamin d activity in ibd”. I recommend discussing cytokines and vitamin D in the setting of IBD as a separate section instead of as a subsection under section 5.
8. The title for section 6 “Vitamin in course of ibd” is confusing and should be revised to better capture the content discussed in section 6.
9. Line 559-577: you described how wnt beta-catenin signaling pathway is involved in bone metabolism regulation, but this was not discussed in relation to IBD. This paragraph should be revised and expanded by making it more relevant to the IBD patient population.
10. Line 639-657: you discussed vitamin d deficiency in IBD patients here in section 9. However, you also discussed this in section 3.3, and some of the content was redundant and should be removed.
11. This review article can be enhanced by adding literature summary tables listing important studies/trials that the authors wish to highlight.
Reviewer 4 Report
I read the manuscript "Pleiotropic effect of vitamin D in patients with inflammatory 2bowel diseases " with great interest. The review of the vitamin D role in inflammatory bowel disease is of considerable interest, and it is a well-written and scientifically sound manuscript. Therefore, I have only a few minor comments:
On some occasion the intestinal microbiota is referred as Intestinal microflora (p.e line 50) this term is not completely correct so; it should be changed.
At point 3.3 Vitamin D deficiency in general population and in inflammatory bowel disease patients, the authors should mention why they consider low levels of vitamin D as < 20ng/ml. The limit of physiological vitamin D concentration is not well stablished, maybe the authors should comment this in this point previously to analyze the levels of vitamin D on IBD patients. The authors consider this in the section 9. Vitamin d and supplementation in patients with IBD but in my opining it is more useful for the reader have this consideration at the beginning of the article.
In section 4 inflammatory bowel disease The references according the incidence of the disease should be updated.
In addition to this there are some typing mistakes p. e in line 222 please check the tying mistake before submit the final version.
Round 2
Reviewer 2 Report
The authors have extensively revised and improved their manuscript. I have a few minor comments:
1. Since section 2 and 3 are combined, section 4 should now be labeled as section 3, section 5 should be labeled as section 4 and so on.
2. Please make sure that all typos and grammatical errors are corrected.
